# Standardizing and Classifying Anterior Cruciate Ligament Injuries: An International Multicenter Study Using a Mobile Application

**DOI:** 10.3390/diagnostics15010019

**Published:** 2024-12-25

**Authors:** Nadia Karina Portillo-Ortíz, Luis Raúl Sigala-González, Iván René Ramos-Moctezuma, Brenda Lizeth Bermúdez Bencomo, Brissa Aylin Gomez Salgado, Fátima Cristal Ovalle Arias, Irene Leal-Berumen, Edmundo Berumen-Nafarrate

**Affiliations:** 1Faculty of Medicine and Biomedical Sciences, Universidad Autónoma de Chihuahua (UACH), Chihuahua 31125, Mexico; kary.portillo15@gmail.com (N.K.P.-O.); sigala.lrsg@gmail.com (L.R.S.-G.); ramos.irrm97@gmail.com (I.R.R.-M.); lizethb2306@gmail.com (B.L.B.B.); gomezaylin0120@gmail.com (B.A.G.S.); cristalovalle@hotmail.com (F.C.O.A.); ileal@uach.mx (I.L.-B.); 2Star Medica Chihuahua Hospital, Perif. de la Juventud 6103, Fracc. El Saucito, Chihuahua 31110, Mexico; 3Faculty of Medicine and Biomedical Sciences, Universidad Autónoma de Chihuahua (UACH), Calle de la Llave #1419 consultorio 9, Colonia Santa Rita, Chihuahua 31020, Mexico

**Keywords:** anterior cruciate ligament (ACL) injuries, pivot-shift test, mobile application, rotational laxity

## Abstract

**Background/Objectives**: This international multicenter study aimed to assess the effectiveness of the Pivot-Shift Meter (PSM) mobile application in diagnosing and classifying anterior cruciate ligament (ACL) injuries, emphasizing the need for standardization to improve diagnostic precision and treatment outcomes. **Methods**: ACL evaluations were conducted by eight experienced orthopedic surgeons across five Latin American countries (Bolivia, Chile, Colombia, Ecuador, and Mexico). The PSM app utilized smartphone gyroscopes and accelerometers to standardize the pivot-shift test. Data analysis from 224 control tests and 399 standardized tests included non-parametric statistical methods, such as the Mann–Whitney U test for group comparisons and chi-square tests for categorical associations, alongside neural network modeling for injury grade classification. **Results**: Statistical analysis demonstrated significant differences between standardized and control tests, confirming the effectiveness of the standardization. The neural network model achieved high classification accuracy (94.7%), with precision, recall, and F1 scores exceeding 90%. Receiver Operating Characteristic (ROC) analysis yielded an area under the curve of 0.80, indicating reliable diagnostic accuracy. **Conclusions**: The PSM mobile application, combined with standardized pivot-shift techniques, is a reliable tool for diagnosing and classifying ACL injuries. Its high performance in predicting injury grades makes it a valuable addition to clinical practice for enhancing diagnostic precision and informing treatment planning.

## 1. Introduction

The knee joint is a complex structure that serves as a pivotal component for weight-bearing and movement. It is primarily composed of three bones, the femur, tibia, and patella, which are supported by a network of ligaments and tendons. These ligaments are critical for maintaining joint stability and include the anterior cruciate ligament (ACL), posterior cruciate ligament (PCL), medial collateral ligament (MCL), and lateral collateral ligament (LCL). The ACL in particular plays a key role in preventing the anterior translation of the tibia relative to the femur and resisting rotational forces. Knee ligament injuries are among the most common musculoskeletal injuries, particularly in active individuals. Injuries often occur due to a sudden axial load on the knee, combined with valgus stress and rotational force around the tibia, and they frequently present with concomitant injuries to the menisci, cartilage, or other knee ligaments [1,2]. Accurately diagnosing these substantial associated injuries is essential. Anterior cruciate ligament (ACL) ruptures should be suspected when the injury mechanism involves deceleration or acceleration combined with valgus loading, along with symptoms such as hearing or feeling a “pop” at the time of injury or hemarthrosis within two hours of the incident.

Clinical evaluation typically includes the Lachman test, which is considered the most accurate diagnostic test for ACL ruptures, with a combined sensitivity of 85% and specificity of 94%. The anterior drawer test is similarly effective in chronic ACL injuries (92% sensitivity, 91% specificity), although it is less precise in acute cases. The pivot-shift (PS) test, regarded as the “gold standard” for assessing rotational knee laxity, has a specificity of 98% for ACL ruptures when positive, but its sensitivity is only 24%, meaning a negative test cannot exclude the injury. Diagnosing a recently injured knee with hemarthrosis often poses challenges due to pain, swelling, and guarding [2,3].

Various imaging modalities are employed to evaluate knee pathology, including X-rays, ultrasonography, computed tomography (CT), and magnetic resonance imaging (MRI). While X-rays are primarily used for initial screening and detecting subtle fractures, CT is better suited for complex fractures but has limited utility for intra-articular pathologies. MRI has emerged as the gold standard for preoperative planning in knee injuries, allowing the classification of ACL ruptures as partial or complete and identifying isolated versus combined injuries [4,5,6].

Despite advances in clinical and imaging techniques, the current methods for evaluating knee stability remain limited. Stability tests and radiographic assessments have an approximate accuracy of 70%. These are supplemented by patient-reported outcome measures such as the International Knee Documentation Committee (IKDC) questionnaire and classifications based on injury chronicity (acute vs. chronic), though their subjective nature often raises concerns about reliability. DeFranco and Bach emphasized the difficulty of diagnosing partial ACL tears, which require a combination of clinical examination, knee laxity evaluation, and arthroscopic assessment [7,8,9,10].

The pivot-shift test, which evaluates dynamic and rotational laxity, is one of the most clinically specific tools for assessing ACL deficiency. This test correlates with subjective instability, reduced sports activity, and meniscal and joint damage. The phenomenon involves anterior tibial translation followed by the posterior reduction of the lateral tibial compartment during multiplanar motion, replicating dynamic knee laxity. However, this test is associated with significant variability due to its reliance on examiner experience and sensitivity. Issues such as inter-observer technique differences, patient apprehension, and variability in IKDC classification have been well documented. As early as 1991, Noyes et al. reported that the pivot-shift test’s classification was imprecise and irreproducible, cautioning its use [11,12,13]. Despite these limitations, the pivot-shift test remains a cornerstone for knee laxity assessment and is a critical component in studies evaluating ACL injuries. High-grade preoperative pivot-shift results have been linked to increased risks of graft failure, persistent instability, and poorer patient-reported outcomes, underscoring the need for more precise quantification methods.

Considering these challenges, efforts to quantify the pivot-shift test and correlate bone movements with injury severity have gained traction. Systems like the KT1000 and KT2000 arthrometers (MED metric Corp., San Diego, CA, USA) quantitatively document sagittal laxity, while rotational laxity can be measured using ambulatory devices (GBRB or ROTAB from the Genourob society), instrumented boots, magnetic resonance imaging, electromagnetic sensors, robotic technology, or navigation systems. Despite significant technological advancements, these tests may not be suitable in a clinical setting, considering issues related to feasibility, affordability, and comfort [14,15,16,17,18].

A smartphone application, the Pivot-Shift Meter (PSM), was developed to address these concerns. By utilizing gyroscopes and accelerometers integrated into mobile phones, the PSM application enables the quantitative analysis of knee rotational instability. Initial comparisons with the KT-1000 arthrometer demonstrated its validity, with 95% of lesion cases classified and inter-observer reliability of 70% in velocity amplitude and 95% in time [19,20].

However, the variability among evaluators in the pivot-shift maneuver remains a challenge. These limitations motivated the current multicenter study, which aims to analyze, classify, and standardize over 400 tests conducted by specialist physicians across Bolivia, Chile, Colombia, Ecuador, and Mexico using the PSM application. By addressing the quantification and standardization of the pivot-shift test, the application offers several potential benefits, such as the ability to classify ACL injuries into severity grades, helping practitioners tailor treatment plans, such as distinguishing between surgical and conservative management options. Quantitative data also aid in surgical planning and rehabilitation monitoring. Unlike expensive and cumbersome devices such as the KT-1000 or KT-2000 arthrometers, the PSM application is portable, cost-effective, and compatible with widely available smartphone technology, making it accessible for use in diverse healthcare settings, including resource-limited environments. By standardizing the pivot-shift maneuver, the PSM app reduces variability among practitioners, ensuring consistency in evaluations and fostering a more reliable understanding of knee laxity. This study seeks to provide a more precise and reliable evaluation method for rotational instability, which is critical for the diagnosis, treatment planning, and postoperative monitoring of ACL injuries.

## 2. Materials and Methods

This multicenter study employed a prospective cohort design to evaluate the effectiveness of the Pivot-Shift Meter (PSM) application in the detection and classification of ACL injuries. The primary objective was to assess the application’s ability to standardize the pivot-shift maneuver and provide objective, quantitative measurements of knee rotational laxity. The trial version of the PSM application served as the primary measurement instrument. The application utilizes inertial sensors (gyroscope and accelerometer) integrated into smartphones to detect and measure ligament injuries in the knee. It provides quantitative data on rotational laxity by analyzing angular velocity in three axes (X, Y, and Z).

This study included both control tests and standardized tests to evaluate the performance of the PSM application:Control Tests: Control tests were conducted prior to the implementation of the standardized pivot-shift app maneuver. These tests relied on traditional clinical evaluation with the use of the PSM application and served as a baseline for comparison.Standardized Tests: After establishing a standardized protocol for the pivot-shift maneuver, tests were performed using the PSM application. The standardized tests were divided as preoperative evaluations conducted on patients scheduled for ACL reconstruction and clinical office evaluations performed to assess knee stability in outpatient settings.

The results from the control and standardized tests were analyzed to demonstrate the improvements achieved through standardization and the use of the PSM application.

### 2.1. Procedure

#### 2.1.1. Participant Recruitment

Specialized orthopedic surgeons from different countries in Latin America, treating a significant number of patients with ACL injuries and experienced in arthroscopic reconstruction, were contacted. A total of 47 healthcare professionals responded to the survey for evaluator recruitment. To ensure reliability and consistency in the study results, specific inclusion criteria were applied, resulting in the selection of 8 orthopedic and trauma specialists. The selection criteria were as follows:Evaluators were required to have extensive experience in diagnosing and treating ACL injuries. An annual average of at least 100 arthroscopic ACL reconstructions was mandatory to ensure proficiency.Commitment to the study protocols, including data collection, accurate application use, and patient follow-up.Training availability for the proper use of the PSM application and the standardized pivot-shift maneuver.

A total of 8 surgeons from Latin America, including Bolivia, Chile, Colombia, Ecuador, and Mexico, agreed to participate in the study and use the PSM application on patients with ligament injuries. Additionally, each patient’s record included information such as gender, age, height, and weight, and the examiner had the facility to add observations, clinical classification according to IKDC criteria, results of digital arthroscopy (KT-1000), results of diagnostic studies, arthroscopy images, and notes with relevant clinical case information.

#### 2.1.2. Sample of the Study

Patients diagnosed with ACL knee injuries were assessed by Latin American orthopedic specialists trained in using the trial version of the PSM application. The study’s inclusion period spanned from February 2022 to December 2023.

##### Inclusion Criteria

Individuals aged 18 years or older.Diagnosed with a primary ACL injury confirmed by a specialist in orthopedic trauma or sports medicine.Individuals scheduled for surgical ACL reconstruction.Provided informed consent to participate.

##### Exclusion Criteria

Individuals under 18 years of age.Individuals with neurovascular injuries or pathologies that may affect study results.Individuals with fractures, re-ruptures, neurological, or muscular problems that impede the performance of the standardized pivot maneuver.

#### 2.1.3. Measurement Procedure/Data Collection

Each surgeon was provided with a guided tutorial to familiarize themselves with the application and an adjustable elastic band to secure their mobile phones, which need to have an accelerometer and gyroscope, with either Android or iOS operating systems. Evaluators used their personal mobile devices to perform tests with the application. Below is a detailed description of the practical steps involved in using the application:Device Setup: The evaluator securely attaches a smartphone with the PSM application installed to the patient’s leg using an adjustable elastic band.

The smartphone is positioned on the tibial tuberosity, approximately two fingers below the patella, and slightly inclined toward the medial aspect of the tibia.

Standardized Maneuver Execution: The evaluator grasps the patient’s ankle medially with the hand corresponding to the side being evaluated, while the opposite hand holds the posterior aspect of the leg at the tibial head. The evaluator applies a slight medial rotation to the tibia and flexes the knee to a 90-degree angle.Data Recording: The PSM application captures real-time angular velocity data along three axes (X, Y, Z) during the pivot-shift maneuver. Each evaluator performs two maneuvers for each patient and saves the results using the application. The application also allows evaluators to add observations, IKDC clinical classifications, results from digital arthroscopy (e.g., KT-1000), diagnostic imaging, arthroscopy images, and other relevant clinical notes.Data Upload and Analysis: The collected data are automatically uploaded to a cloud-based database for further analysis. The application processes the data using neural network algorithms, classifying the degree of rotational laxity and providing immediate feedback to the evaluator.

#### 2.1.4. Data Interpretation

Statistical analyses were performed to evaluate the effectiveness of the PSM application in assessing and classifying ACL injuries. The decision was made to analyze the *X*-axis of the gyroscope since it reflected the flexion and extension of the knee during the pivot-shift maneuver. Features were extracted from the raw *X*-axis data vector of all evaluators and subsequently processed using a program designed in MATLAB. The following methods were used:Mann–Whitney U Test: Employed to compare results between control tests and standardized tests, as the data did not meet normal distribution assumptions.Chi-Square Test (χ^2^): Used to analyze the association between control and standardized test distributions, providing statistical evidence of the impact of the standardized pivot-shift maneuver on test classifications.ROC Curve and Area Under the Curve (AUC) Analysis: Conducted to evaluate the diagnostic accuracy of the PSM application.

### 2.2. Variables and Outcomes

This study focused on the following variables and outcomes to evaluate the effectiveness of the PSM app in ACL injury diagnosis and classification:Rotational Laxity: Measured during the pivot-shift maneuver using angular velocity data captured by the PSM application along three axes (X, Y, Z).Injury Classification: Grades of ACL injuries were classified using a neural network model trained on data collected from the pivot-shift maneuver and validated against IKDC criteria.

#### Outcomes of Interest

Neural Network Performance Metrics: Accuracy, precision, recall, and F1-score in classifying ACL injuries.Diagnostic Accuracy: Receiver Operating Characteristic (ROC) analysis and Area Under the Curve (AUC) values to assess the model’s ability to differentiate injury grades.Impact of Standardization: Comparison of standardized versus control tests to evaluate the improvements in diagnostic reliability and consistency achieved through the standardized pivot-shift technique.

## 3. Results

### 3.1. Overview of Tests Conducted

In total, 437 tests were captured using the standardized pivot method, with 233 tests on the right knee and 204 on the left knee. The results in all three axes of movement (X, Y, and Z) were obtained using the gyroscopes integrated into the evaluator’s smartphone (Figure 1). Among these, 136 tests were conducted in a medical office due to clinical evidence of ACL injury during physical evaluation, 180 tests were performed during preoperative evaluation in patients diagnosed with ACL injury scheduled for arthroscopic reconstruction, 63 tests were conducted during immediate postoperative evaluation in postoperative patients, and 20 tests were performed in patients with a history of arthroscopic reconstruction to track their progress. Out of the 437 registered tests, 38 were discarded as they did not meet the standards for analysis. These included tests with technical failures during execution, incorrect placement of the mobile device, or incorrect execution of the standardized pivot maneuver, leaving a total of 399 standardized tests (Figure 2). According to the examiners, 60% of ACL injuries were classified as Grade I, 33% as Grade II, and 7% as Grade III, based on the IKDC criteria. Subsequently, the information from the database was downloaded and organized into a comma-separated values (.csv) file using a Python-generated code for further analysis.

Once the signal was acquired, the *X*-axis gyroscope data were divided into three sections. Mathematical, morphological, and statistical characteristics of four types of data representation were then analyzed: the original values, normalized values, Fourier transform of the original values, and Fourier transform of the normalized values (Figure 3).

After examining and evaluating the different features and data representations, it was concluded that segment 3 was the most relevant and representative for capturing the amplitude or magnitude of the knee movement response in the *X*-axis during the pivot-shift maneuver. Consequently, this segment was used to categorize the tests into different classes. The following mathematical characteristics of segment 3 were taken into account for this categorization: standard deviation (S3n-STD), range (S3n-R), range plus the average of the segment (S3n-Rx), the ordinate of the linear regression (S3n-O), and the slope of the linear regression (S3TF-P).

Following the exclusion of tests with failures and obtaining the characteristics of standardized tests, the analysis for their reclassification was performed. The results of the multicenter study, consisting of 399 tests, were distributed into 8 classes.

### 3.2. Statistical Analysis Results

#### Association Between Control and Standardized Tests

Results of 224 control tests obtained using the first version of the PSM application, before the implementation of standardization in the pivot technique, were also included. These tests were distributed into their respective classes (Table 1).

Statistical analysis results revealed that the calculated value for the χ^2^ test was 62.17213413. When compared to the critical value of 14.067, with a significance level of 0.05, a *p*-value of 0.00001 was obtained, indicating a significant association between control tests and standardized tests. These findings support the effectiveness of standardizing the pivot technique, as it has had a notable impact on the distribution of the classes.

### 3.3. Neural Network Performance

With standardized tests and control tests reclassified into eight classes, the obtained characteristics described above were used to create a neural network for class groups. The network was trained with the results recorded in the application, and a general performance evaluation of the model in the classification task was performed (Figure 4; Table 2).

The performance analysis results of the created model revealed highly favorable performance in terms of its ability to classify maneuvers into the desired class. Accuracy, precision, and sensitivity indicators exceeded 90%, demonstrating the model’s high capability to accurately classify the maneuver class. Specifically, the F1 score of the model reached a result of 97.27%, confirming its overall accuracy and sensitivity in classifying the classes. These results support the effectiveness and reliability of the model in the objective identification and classification of standardized maneuvers.

#### Diagnostic Accuracy

Next, additional analysis was conducted to evaluate the ability of each class to assess the degree of ACL injury in four levels: grades 0, 1, 2, and 3.

For this, a confusion matrix specific to the classes was created to evaluate the classification function of the injury according to its corresponding grade. The following mathematical, statistical, and morphological characteristics of significant signals for grade assignment by class were analyzed: maximum value of segments S1 and S3 (S1-Max, S3-Max), ranges of segments S2 and S3 (S2-R, S3-R), standard deviation of S3 (S3-STD), Shannon entropy of S3 (S3-We), and the slope of the linear regression of segment 3 (S3TF-P). During the analysis, it was observed that classes 1 and 8 represented maneuvers furthest from the standardized method, so their classification results by grade were discarded.

The following results were reported for classes 2 to 7 (Figure 5; Table 3).

The class-specific analysis yielded highly favorable results in terms of neural network performance indicators. In all analyzed classes, accuracy, precision, and sensitivity exceeded 90%, demonstrating the model’s ability to accurately classify ACL injuries into different grades, according to the PSM maneuver class. The F1 score was calculated, and notable results were obtained. In Class 2, an F1 score of 98% was achieved, indicating the model’s high ability to classify injury grade, particularly for this specific category of PSM maneuver. In Class 3, an F1 score of 97% was achieved, and in Classes 4, 5, 6, and 7, F1 scores of 96%, 94%, 95%, and 96%, respectively, were recorded.

These results confirm the software’s ability to classify with high accuracy and reliability the grade of ACL injury by analyzing the amplitude or magnitude of the knee movement response during the pivot-shift maneuver, as represented in segment 3 of the X-axis of the gyroscope data.

The evaluation of the performance of the neural network model in classifying standardized maneuvers and in determining the degree of ACL injury according to grade clearly demonstrates its high efficiency, sensitivity, and reliability. The analysis of variance was carried out using the Shapiro–Wilk test, which indicated that the sample was not distributed normally. This was confirmed by the Bartlett test, which rejected the hypothesis of homoscedasticity. Due to these findings, non-parametric tests were chosen for the analysis of the data.

To determine the statistical significance of the differences between the results obtained for standardized tests and control tests, the Mann–Whitney U test was applied, which confirmed the presence of statistically significant differences in the results obtained between the groups.

To evaluate the accuracy of the device in diagnosing ACL injuries, the ROC curve was analyzed using the results of standardized tests and control tests, which enabled the calculation of the AUC. The ROC curve reflects the ability of the device to differentiate between ACL injuries and non-injuries, with an AUC of 0.80 indicating a high accuracy rate.

### 3.4. Key Takeaways

Statistical analysis showed a significant association between control and standardized test distributions (χ^2^ = 62.17, *p* < 0.00001), supporting the effectiveness of the standardized pivot-shift maneuver in improving test reliability.

The neural network model developed for injury classification achieved high performance, with

Accuracy: 94.7%;Precision: 98.0%;Recall: 96.6%;F1-Score: 97.3%.

Classification accuracy exceeded 90% for all injury grades analyzed (Grades I–III).

These results support the PSM application’s utility as a reliable tool for standardizing the pivot-shift maneuver, improving ACL injury classification, and enhancing diagnostic precision.

## 4. Discussion

This study evaluated the effectiveness of the PSM app in standardizing the pivot-shift maneuver and classifying ACL injuries, structured around two primary objectives: assessing the impact of the PSM application on standardizing the pivot-shift test and evaluating the application’s accuracy in classifying ACL injuries.

The pivot-shift test is considered a cornerstone for assessing rotational knee laxity but has historically been limited by inter- and intra-observer variability [11,12]. The results of this study demonstrate that the PSM application significantly reduced this variability through a standardized protocol, as evidenced by the χ^2^ test results (χ^2^ = 62.17, *p* < 0.00001). These findings support the utility of the PSM application in addressing challenges noted in earlier studies by providing consistent and reproducible measurements of knee laxity during the pivot-shift maneuver [21,22,23,24,25].

A major contribution of this study is the use of a neural network model to classify ACL injuries based on gyroscope and accelerometer data. The classification accuracy of 94.7%, with precision, recall, and F1-scores exceeding 90%, underscores the robustness of the PSM application as a diagnostic tool. Compared to traditional devices like the KT-1000, which focus on sagittal laxity, the PSM application offers a more comprehensive evaluation by quantifying rotational instability, a critical parameter for assessing ACL deficiencies [26,27,28,29,30,31,32].

The PSM application stands out as a portable and cost-effective alternative to devices such as the KT-2000 and intraoperative optical navigation systems [30,31,32,33,34]. These methods, while precise, are either invasive or not feasible in routine clinical settings. By leveraging smartphone gyroscopes and accelerometers, the PSM application addresses the need for accessible diagnostic tools in diverse clinical environments [11,12]. Its compatibility with Android and iOS platforms further enhances its practicality, particularly in resource-limited healthcare settings.

The ability of the PSM application to classify ACL injuries into grades (0, I, II, and III) has significant implications for treatment planning. This classification, validated against IKDC criteria, provides clinicians with a detailed understanding of injury severity. Accurate grading is crucial for selecting appropriate surgical techniques, particularly when considering the discrepancies in outcomes between single-bundle and double-bundle ACL reconstructions reported in the literature [7,9,22]. Furthermore, the continuous data generated by the PSM application offers a more nuanced assessment compared to categorical IKDC grades, addressing concerns about subjectivity and variability in clinical evaluations [13,26,27,28].

We acknowledge several limitations in this study. First, although the multicenter design enhances the generalizability of our findings, the sample size was limited to a specific population of Latin American patients. This may limit the applicability of our results to other regions or demographic groups. Second, despite the standardized procedures implemented, there remains some potential for variability in the execution of the pivot-shift test, as the test’s reliability is influenced by the evaluator’s experience. Additionally, while the PSM application significantly reduced inter-observer variability, some technical issues, such as improper device placement, were observed in a small number of cases. Finally, the study did not track long-term outcomes, such as the recovery progress or functional improvements post-surgery, which could further validate the PSM application’s utility in clinical practice.

To further validate the PSM application, longitudinal studies are needed to evaluate its performance in monitoring postoperative recovery and functional improvements. Additionally, direct comparisons with advanced arthrometers could establish its role as a gold standard for ACL injury evaluation. These advancements will enhance the PSM application’s potential to transform ACL injury diagnosis and management.

## 5. Conclusions

This study demonstrates the effectiveness of the PSM app in standardizing the pivot-shift maneuver and classifying ACL injuries with high accuracy. By leveraging smartphone-based inertial sensors, the PSM application provides a cost-effective, accessible, and reliable tool for diagnosing and quantitatively assessing rotational knee instability.

The use of a standardized maneuver significantly improved the consistency and reliability of test classifications, while the neural network model achieved over 94% accuracy in predicting injury grades, confirming the study’s hypothesis. These findings highlight the potential of integrating smartphone technology into clinical practice to enhance diagnostic precision and treatment planning for ACL injuries.

Future work should focus on validating the PSM application in longitudinal studies to evaluate its utility in tracking recovery and functional outcomes. Additionally, direct comparisons with advanced arthrometers will further establish its role as a standard tool for ACL injury assessment.

## Figures and Tables

**Figure 1 diagnostics-15-00019-f001:**
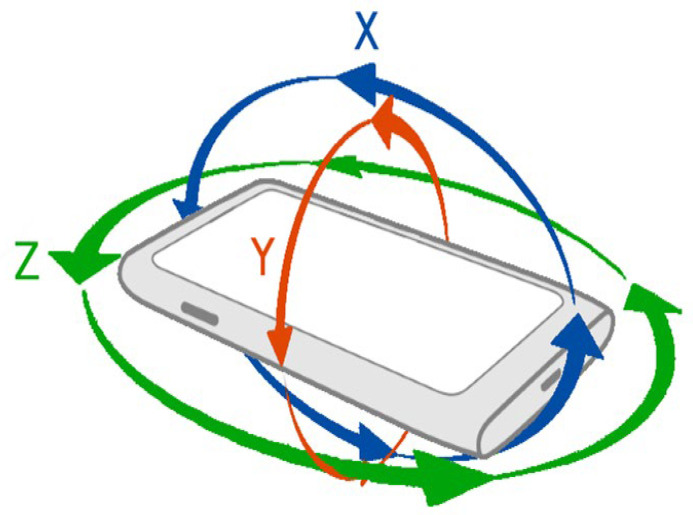
Cellphone axes of rotation. Each axis corresponds to the rotation in one direction of the device. The arrows represent the X, Y, and Z axes, indicating the rotational planes of the gyroscope used during the pivot-shift maneuver.

**Figure 2 diagnostics-15-00019-f002:**
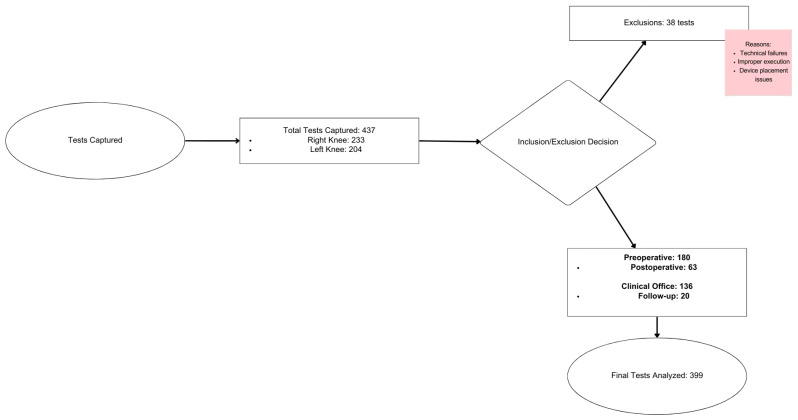
Tests performed flow chart.

**Figure 3 diagnostics-15-00019-f003:**
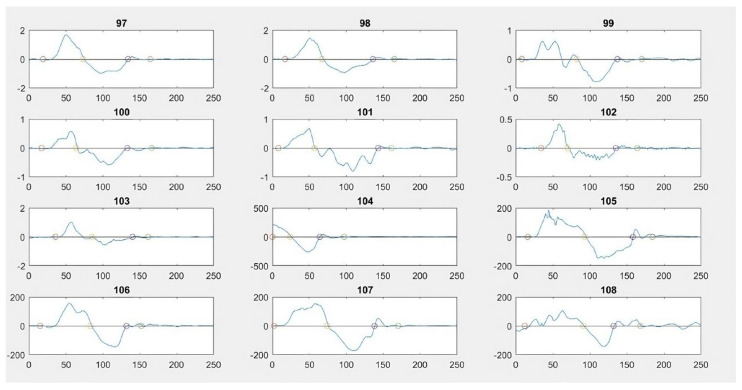
Results of tests from one of the evaluators on 12 different patients. The signal is divided into three segments, S1, S2, and S3, in each maneuver.

**Figure 4 diagnostics-15-00019-f004:**
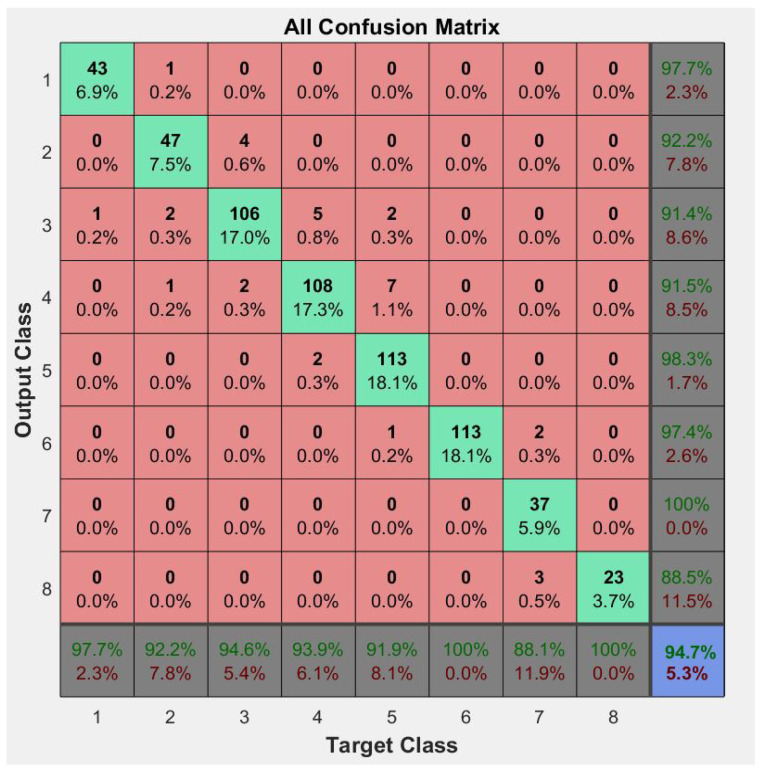
Confusion matrix results of the class classification model.

**Figure 5 diagnostics-15-00019-f005:**
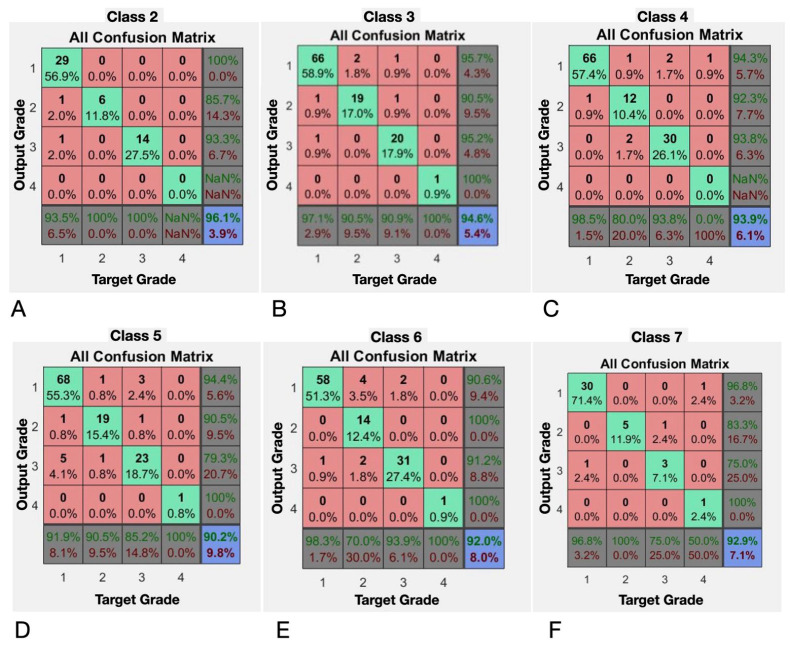
Confusion matrices for neural network performance by class (**A**–**F**), representing classification accuracy for ACL injury grades across Classes 2 to 7.

**Table 1 diagnostics-15-00019-t001:** Observed frequency contingency table: control tests and standardized tests.

Class	Control Tests	Standardized Tests	Total Tests
1	33	11	44
2	23	28	51
3	31	81	112
4	36	79	115
5	33	90	123
6	30	83	113
7	21	21	42
8	17	6	23
Total	224	399	623

**Table 2 diagnostics-15-00019-t002:** Class and metrics results for neural network evaluation.

Class	True Positive	False Positive	False Negative	Total	Accuracy	Precision	Recall	F1-Score
1	43	1	0	44	97.727	97.72727273	100	98.8505747
2	47	3	1	51	92.2	94	97.91666667	95.9183673
3	106	2	4	112	94.6	98.14814815	96.36363636	97.2477064
4	108	2	5	115	93.9	98.18181818	95.57522124	96.8609865
5	113	1	9	123	91.9	99.12280702	92.62295082	95.7627119
6	113	0	0	113	100.0	100	100	100
7	37	3	2	42	88.1	92.5	94.87179487	93.6708861
8	23	0	0	23	100.0	100	100	100
Total	590	12	21	623	94.7	98.00664452	96.56301146	97.2794724

**Table 3 diagnostics-15-00019-t003:** Average neural network performance metrics on grade classification tasks: the accuracy, precision, recall, and F1-score are shown for each class (2 to 7), highlighting the model’s high ability to accurately classify the grades of ACL injuries.

Evaluated Class	Accuracy%	Precision%	Recall%	F1-Score%
2	96.1	96.0784314	100	98
3	94.6	98.1481481	96.3636364	97.2477064
4	93.9	97.2972973	96.4285714	96.8609865
5	90.2	94.0677966	95.6896552	94.8717949
6	92	97.1962617	94.5454545	95.8525346
7	92.9	97.5	95.1219512	96.2962963
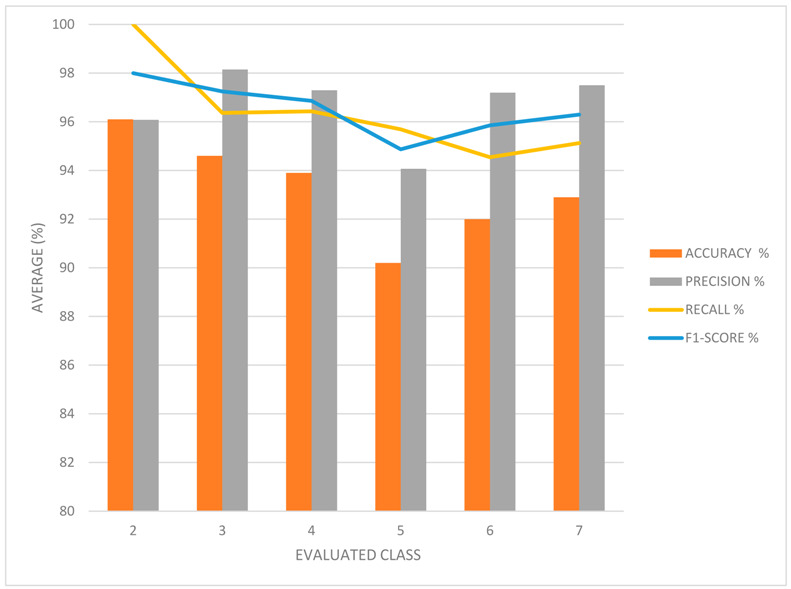				

## Data Availability

The data that support the findings of this study are available from the corresponding author upon reasonable request.

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
