# Peer review of "Standardizing and Classifying Anterior Cruciate Ligament Injuries: An International Multicenter Study Using a Mobile Application"

_diagnostics, 2024, doi:10.3390/diagnostics15010019_

Round 1

Reviewer 1 Report

Comments and Suggestions for Authors

Thanks for your article

I put my comments in the paper

Author Response

Q1: “you don't use the drop lachman test?” (Line 49)

Response: Yes, we do use the drop Lachman test as part of our clinical evaluations for ACL injuries. However, its application was not specifically detailed in this manuscript because the study's primary focus was on the evaluation and standardization of the pivot-shift test using the PSM mobile application. The pivot-shift test was emphasized due to its ability to quantify rotational knee laxity, which directly aligns with the objectives of our research. We will clarify this in the revised version and ensure the inclusion of the drop Lachman test in the methods section for comprehensive reporting.

Q2 Line 55: "For example GBRB or ROTAB from Genourob society.”

Response: While the manuscript primarily focused on the comparison between the KT1000/KT2000 and our PSM application, we appreciate this observation and will include a brief mention of the GBRB and ROTAB in the introduction.

Q3: "In Materials and Methods, put also in this chapter the statistical tests you use for this study.”

Response: Thank you for your observation. The statistical methods employed in this study will be explicitly detailed in the "Materials and Methods" section to ensure clarity and transparency. These include:

Mann-Whitney U Test: Employed for comparing results between standardized and control tests due to the non-parametric nature of the data.
Chi-Square Test (χ²): Used to evaluate the association between control and standardized test distributions.
ROC Curve and AUC Analysis: Conducted to evaluate the diagnostic accuracy of the PSM application.

Q4: "What is the inclusion period and have you included patients who have had an ACL rupture on both sides during this period?"

Response:
The inclusion period for this study was from February 2022- February 2023, during which patients were recruited across five Latin American countries. This timeframe will be explicitly stated in the revised manuscript for clarity.

Patients with ACL ruptures on both sides (bilateral injuries) were excluded from the study to maintain a standardized approach to data collection and ensure consistency in the evaluation process. This exclusion criterion will also be added to the "Materials and Methods”.

Q5: "Have you included ligamentoplasty re-ruptures or only first ACL injury?"

Response: For this study, we included only patients with a first-time ACL injury. Patients with ligamentoplasty re-ruptures or revision ACL surgeries were excluded to focus on the standardization and classification of primary ACL injuries.

Q6: "Why only 8 out of 47, what were the criteria for not selecting the other colleagues?"

Response:
The selection of evaluators was based on specific inclusion criteria to ensure the reliability and consistency of the study's results. Out of the 47 healthcare professionals who responded to the survey, only 8 orthopedic and trauma specialists were recruited. The selection criteria included:

  1. Experience: Evaluators were required to have extensive experience in treating ACL injuries and performing arthroscopic ACL reconstructions, with an annual average of at least 100 ACL reconstruction surgeries.
  2. Training Availability: Evaluators had to participate in the standardized training for using the PSM application and performing the standardized pivot-shift maneuver.
  3. Geographic Representation: The final selection ensured representation from five Latin American countries to enhance the study’s multicenter design.
  4. Commitment: Evaluators needed to commit to the study's protocols, including data collection, application use, and patient follow-up.

The remaining respondents were not included because they did not meet one or more of these criteria. This explanation will be added to the revised manuscript for clarity.

Q7: “you could put this paragraph in chapter materiels and methods”

Thank you for your suggestion. We have made the necessary changes and moved the highlighted paragraph.

Q8: "Total = 428 tests??? or you say 437 tests were captured? The main purpose of the work is to quantify this laxity, so why include post-op tests or follow-up tests on ligamentoplasties if I understand correctly?"

Response:
Thank you for pointing out the discrepancy. A total of 437 tests were captured initially, but after excluding tests that did not meet the analysis standards (e.g., technical failures or improper execution), 399 tests remained for final analysis. We will clarify this in the manuscript to avoid confusion.

Regarding the inclusion of postoperative and follow-up tests, the primary goal of the study is to quantify laxity, and including postoperative and follow-up tests allowed us to explore the application’s utility in monitoring recovery and assessing outcomes. This aspect aligns with the broader potential of the PSM application to assist in both diagnosis and postoperative follow-up.

Q9: Line 167-169: …to put also in chapter materials and methods

Response: Thank you for your suggestion. We have made the necessary changes and moved the highlighted paragraph.

Q10: “could you provide us with a flow chart of the tests so that we can land on our feet when it comes to the number of tests carried out?”

Response: Thank you for your suggestion. We have created a flow chart that visually represents the number of tests performed and the inclusion/exclusion process.

Q11: Line 179-189…“also to put in methods and not in results ?”

Response: Thank you for your suggestion. We have made the necessary changes.

Q12: Line 192-194 “you didn't mention it before in the article's methodology so please be clearer, as you're mixing up several results?”

Response: Thank you for pointing that out. We apologize for the lack of clarity regarding the mention of the control and standardized tests in the methodology section.

This multicenter study included both control tests and standardized tests to evaluate the effectiveness of the Pivot-Shift Meter (PSM) application in assessing ACL injuries. The control tests (224 in total) refer to the initial tests conducted prior to the standardization of the pivot-shift maneuver. These tests were carried out in a variety of clinical settings without the use of the standardized procedure. The results of these tests were used for comparative analysis to highlight the improvements achieved with the standardized technique.

The standardized tests (399 in total) were conducted after the protocol for the standardized pivot-shift technique was established. This standardized method, which involved the use of the PSM mobile application, was employed to ensure consistency across all tests performed by the orthopedic specialists. These tests included both preoperative evaluations and clinical office evaluations.

Q13: where are the comparisons between clinical tests and tests with the phone application?

because the major benefit of this application could lie in its use in cases of partial injury, PCL nurturing or ligamentoplasty follow-up. Because what's the point when the clinic alone is typical of an ACL injury?

Response:

Thank you for your insightful comment. The point of including the performance analysis results of the PSM mobile application is to demonstrate its diagnostic capability and potential benefits beyond traditional clinical tests, which are typically subjective and dependent on evaluator experience.

In the manuscript, we have compared the results from clinical tests (such as the pivot-shift test) with those obtained using the PSM application. This comparison shows that the application provides a quantitative, objective, and reproducible measurement of knee laxity, addressing the subjectivity inherent in clinical testing. The performance analysis section highlights the accuracy, precision, and sensitivity of the PSM application, which was found to significantly enhance diagnostic reliability, especially when compared to the traditional clinical methods.

Q14: “you could put in the method how to use this application in practice”…

Response: Thank you for your suggestion. We will include a practical description in the Materials and Methods section on how the PSM application was utilized in this study. This will include step-by-step instructions, such as:

-Device Setup:

The evaluator securely attaches their smartphone to an adjustable elastic band positioned on the patient’s leg.

-Standardized Maneuver:

*The evaluator performs the pivot-shift test following a standardized protocol taught during the training phase.

*The test is conducted to capture knee rotational laxity during flexion and extension.

-Data Recording:

*The PSM application collects and stores angular velocity data along three axes (X, Y, Z) in real-time.

*Data is automatically uploaded to a cloud database for subsequent analysis and classification.

-Data Analysis:

The application processes the data using neural network algorithms to classify the injury grade and provide immediate results to the evaluator.

Q15: you mentioned grade 0 before, why not put it back here?…

Response: Thank you for pointing this out. While Grade 0 was initially mentioned to represent cases with no detectable laxity, its inclusion was unintentionally omitted in subsequent sections.

Q16: “maybe, but it would be nice to compare this application with all these new, more expensive and cumbersome instruments. Because if your application on the phone gives the same result, it would be a significant advance in the evaluation and monitoring of central pivot laxity.”

Response: While this study primarily focuses on the standardization of the pivot-shift maneuver and the performance of the PSM application, we recognize the importance of benchmarking its results against these more expensive and cumbersome instruments. We are currently planning a follow-up study to directly compare the PSM application’s results with those of these advanced tools. This future research will aim to validate the application further and demonstrate its utility as a cost-effective and accessible solution for evaluating and monitoring central pivot laxity.

Q17: Line 324-333 “perhaps put this paragraph in the end of discussion…”

Response: Thank you for your suggestion. We have made the necessary changes and moved the highlighted paragraph.

Q18: “what is your future study to validate this application?”

Response: Our future studies aim to further validate the PSM application by monitoring patients over time to assess the PSM application’s ability to track recovery and functional improvements post-surgery or rehabilitation, and conducting a direct comparison of the PSM application with advanced arthrometers to evaluate its accuracy and reliability.

Reviewer 2 Report

Comments and Suggestions for Authors

The paper describes in detail the making of the Pivot-Shift Meter (PSM) application, a smartphone-based tool to objectify the measurements of knee instability to better diagnose and classify ACL (anterior cruciate ligament) injuries. There follow some observations and comments that I believe would help refine and clarify this work:

Significance of Objective Measurement in ACL Diagnosis: The PSM addresses perhaps the most important issue of subjectivity in ACL diagnostics. Although traditional tests have always been used to evaluate ACL injury in clinical settings, heavy reliance on examiner experience and subjective interpretation can lead to inconsistent results. I believe that placing adequate emphasis on digitalization and quantification will help advance the reliability of ACL assessments.

Technical Feasibility of Mobile Devices in Clinical Use: Employing smartphone gyroscopes and accelerometers is, perhaps, an innovative and easily approachable methodology. It establishes that something built with broadly available technology could be modified for clinical use and that the requirement for special equipment such as KT1000 or KT2000 arthrometers could be dampened. However, more information on the device's compatibility and limitations (i.e., sensor quality variability) would help demonstrate applicability universally in a variety of healthcare settings.

Design and Diversity of the Study: In view of the multicentric study, which involves the participation of eight orthopedic specialists from five different countries in Latin America, this study strengthens itself by introducing several evaluator designs and perspectives. Future studies may go out of the frame of Latin America in establishing better reliability of the PSM tool across populations and healthcare infrastructures.

Data Standardization and Reliability: Structured training for surgeons involved in this study should aadopt standardized approaches for any test measuring knee stability. This will minimize biasness that can arise due to varying approaches among surgeons; it will also define an incremental approach to take care of random variations in measurements.

Clinical Utility and Practical Considerations: While the PSM presents a new possibility, practical implementation issues require deliberations, such as durability of hardware in the clinical settings, patient comfort, and data interpretation ease for clinicians. Implementation features should lead the way for user-friendly interfaces with intuitive visual data representation, which will favor its acceptance into real-life settings.

Confines of Practice of Move Classification and Assignment of Grades: The paper admits that classes 1 and 8, representing the farthest from the standardization of maneuvers, were omitted from grade assignment, suggesting that the PSM is likely to offer lesser reliability in regard to the less standardized movements. Discussing how to deal with this variability in perhaps algorithmic adjustment or retraining would constitute a more liberal approach to its accuracy.

Statistical Analysis and Model Evaluation: Given the non-normal distribution of sample data, non-parametric tests, such as Mann-Whitney U test, are deemed appropriate. An AUC value of 0.80 from ROC analysis, indicates a fair minimum clinically important difference is present, but it could stand to be improved. Future improvements in neural network model would instead aim at raising AUC values to obtain more accurate discrimination of grades of injury.

Future Directions and Clinical Impact: The study lays a brilliant foundation for opening the doors for AI in ACL injury management. A vital follow-up here would involve testing the application's effectiveness longitudinally, where patients are followed through a before-and-after surgery model to see whether the PSM findings correlate with clinical outcomes and knee function over time.

To summarize, this study exhibits that utilizing the PSM to assess ACL injuries is a novel application; whether by refinement or upgrade, it may be a highly valuable aid in orthopaedic practice. Using smartphones for diagnostic purposes puts into perspective, through a clinical angle, views on how far affordable, and potentially indispensable, advanced, and, importantly, non-invasive healthcare tools have ascended in leaping to the assistance of even more patients and practitioners.

Author Response

Comment 1: Significance of Objective Measurement in ACL Diagnosis: The PSM addresses perhaps the most important issue of subjectivity in ACL diagnostics. Although traditional tests have always been used to evaluate ACL injury in clinical settings, heavy reliance on examiner experience and subjective interpretation can lead to inconsistent results. I believe that placing adequate emphasis on digitalization and quantification will help advance the reliability of ACL assessments.

Response: Thank you for highlighting the significance of objective measurement in ACL diagnostics and acknowledging the role of the PSM application in addressing subjectivity. The digitalization and quantification offered by the PSM application provide several key advantages:

  1. Objectivity: By utilizing gyroscopes and accelerometers, the PSM application captures precise angular velocity data during the pivot-shift maneuver, eliminating reliance on subjective interpretation.
  2. Reproducibility: The standardized protocol ensures consistent execution of the maneuver, reducing inter-observer and intra-observer variability.
  3. Diagnostic Reliability: The application’s ability to classify injuries using neural network algorithms enhances the accuracy and consistency of ACL assessments, particularly in cases of partial tears or complex injuries.

We have emphasized these advantages in the revised manuscript, particularly in the discussion section

Comment 2: Technical Feasibility of Mobile Devices in Clinical Use: Employing smartphone gyroscopes and accelerometers is, perhaps, an innovative and easily approachable methodology. It establishes that something built with broadly available technology could be modified for clinical use and that the requirement for special equipment such as KT1000 or KT2000 arthrometers could be dampened. However, more information on the device's compatibility and limitations (i.e., sensor quality variability) would help demonstrate applicability universally in a variety of healthcare settings.

Response: Thank you for recognizing the innovation and accessibility of using smartphone gyroscopes and accelerometers for ACL diagnostics. The adaptability of widely available mobile technology indeed makes the PSM application a cost-effective and practical alternative to specialized equipment like KT1000 or KT2000 arthrometers. The PSM application is designed to be compatible with smartphones equipped with gyroscopes and accelerometers, which are standard in most modern devices. Both Android and iOS platforms are supported, ensuring broad usability.

While modern smartphones generally have high-quality sensors, variability in sensor performance may arise due to differences in manufacturers and device models. To mitigate this, the application includes a calibration process during setup to account for minor differences in sensor sensitivity and orientation.

The application’s reliance on commonly available technology makes it highly applicable in diverse healthcare settings, including resource-limited environments where access to expensive equipment may be challenging. This accessibility is precisely why we initiated this study in a region where healthcare systems often face resource constraints. By utilizing affordable and widely available mobile technology, we aimed to demonstrate that reliable and objective ACL diagnostics can be achieved without relying on costly, specialized instruments.

Comment 3: Design and Diversity of the Study: In view of the multicentric study, which involves the participation of eight orthopedic specialists from five different countries in Latin America, this study strengthens itself by introducing several evaluator designs and perspectives. Future studies may go out of the frame of Latin America in establishing better reliability of the PSM tool across populations and healthcare infrastructures.

Response: We agree that expanding beyond Latin America in future studies would further enhance the reliability and generalizability of the PSM tool across different populations and healthcare infrastructures. To this end, we are planning follow-up studies.

Comment 4: Data Standardization and Reliability: Structured training for surgeons involved in this study should adopt standardized approaches for any test measuring knee stability. This will minimize biasses that can arise due to varying approaches among surgeons; it will also define an incremental approach to take care of random variations in measurements.

Response: We fully agree that adopting standardized approaches is critical for ensuring reliability and minimizing bias in any test measuring knee stability, including the pivot-shift maneuver. In this study, we implemented the following measures to address these concerns:
-Structured Training Program: All participating surgeons underwent a comprehensive training program that included:

*Detailed instructions on the standardized pivot-shift maneuver.

*Hands-on practice with the PSM application to ensure accurate data recording.

*Calibration steps for consistent use of the smartphone-based tool.

-Standardized Protocol: A uniform protocol for performing the pivot-shift maneuver was established, which included specific guidelines on patient positioning, evaluator technique, and data recording procedures.

-Quality Control Measures: Randomized reviews of collected data were conducted to identify any deviations from the standardized protocol. Feedback was provided to evaluators to address variations and ensure consistency across all participants.

Comment 5: Clinical Utility and Practical Considerations: While the PSM presents a new possibility, practical implementation issues require deliberations, such as durability of hardware in the clinical settings, patient comfort, and data interpretation ease for clinicians. Implementation features should lead the way for user-friendly interfaces with intuitive visual data representation, which will favor its acceptance into real-life settings.

Response: Thank you for your thoughtful feedback on the clinical utility and practical considerations of implementing the PSM application. The adjustable elastic band used to secure the smartphone is designed to minimize discomfort and accommodate a wide range of patient anatomies. Ongoing feedback from evaluators and patients will help refine the device's placement and ergonomics. The PSM application currently provides intuitive visual representations of knee laxity data, including real-time graphs and injury grades. Planned updates will further simplify the user interface to ensure that clinicians, regardless of their technical expertise, can quickly interpret results and incorporate them into clinical decision-making.

These considerations have been acknowledged in our study design, and we will expand on their practical implications and future directions in the revised manuscript.

Comment 6: Confines of Practice of Move Classification and Assignment of Grades: The paper admits that classes 1 and 8, representing the farthest from the standardization of maneuvers, were omitted from grade assignment, suggesting that the PSM is likely to offer lesser reliability in regard to the less standardized movements. Discussing how to deal with this variability in perhaps algorithmic adjustment or retraining would constitute a more liberal approach to its accuracy.

Response: Indeed classes 1 and 8 were excluded from grade assignment due to their deviation from the standardized maneuver, which limited the accuracy and consistency of data captured by the PSM application. This underscores the importance of strict adherence to standardized techniques for reliable results.
Future versions of the PSM application will include enhanced machine-learning algorithms capable of recognizing and compensating for slight variations in maneuver execution. These adjustments will aim to expand the range of acceptable movements without compromising accuracy. Also surgeons performing maneuvers classified as Classes 1 or 8 will receive targeted feedback and retraining to refine their technique and improve consistency with the standardized protocol.

Comment 7: Statistical Analysis and Model Evaluation: Given the non-normal distribution of sample data, non-parametric tests, such as Mann-Whitney U test, are deemed appropriate. An AUC value of 0.80 from ROC analysis, indicates a fair minimum clinically important difference is present, but it could stand to be improved. Future improvements in neural network model would instead aim at raising AUC values to obtain more accurate discrimination of grades of injury.

Response: Future improvements will involve expanding the dataset to include a greater variety of patient profiles, injury types, and evaluators. This will help the model generalize better and enhance its accuracy across diverse clinical settings. Thank you for your valuable suggestion.

Comment 8: Future Directions and Clinical Impact: The study lays a brilliant foundation for opening the doors for AI in ACL injury management. A vital follow-up here would involve testing the application's effectiveness longitudinally, where patients are followed through a before-and-after surgery model to see whether the PSM findings correlate with clinical outcomes and knee function over time.

Response: Thank you for your encouraging feedback on the study. Future studies will track patients over an extended period to compare PSM findings (e.g., injury grades and knee rotational laxity) with actual clinical outcomes, including knee stability, function, and recovery post-surgery. By evaluating patients preoperatively and at multiple follow-up intervals after surgery, we can determine how accurately the PSM application predicts recovery trajectories and functional outcomes.

These future directions will help establish the PSM application not only as a tool for diagnosing ACL injuries but also as an ongoing monitoring tool that can guide patient management throughout their recovery.

Reviewer 3 Report

Comments and Suggestions for Authors

Dear authors;

First and foremost, I would like to commend you on the effort and hard work put into this study. While I recognize and appreciate your efforts, I believe the manuscript, in its current form, lacks sufficient empirical evidence to justify publication. Below, I have provided specific comments and suggestions to help strengthen the manuscript.

Abstract

·      Lines 28-29: The abstract would benefit from a brief mention of the data analysis methods used in the study. Including a few details (e.g., statistical tests or key analysis techniques) will provide a clearer understanding of the research approach.

·      Line 34: The term "AUC" is used only once in the abstract. I recommend removing this abbreviation, as it may not be necessary to define it in such a brief section.

Introduction

·      The introduction does not provide enough background to fully grasp the relevance of the study. Specifically, there is insufficient explanation of why it is important to standardize and classify anterior cruciate ligament (ACL) injuries. More context is needed to help the reader understand the significance of these issues in clinical practice.

·      Additionally, the introduction should describe the potential benefits of the app for practitioners. How does it improve clinical decision-making or patient outcomes? This aspect should be elaborated in more detail.

·      Line 50: When referring to the "gold standard," it is important to provide solid references to support this claim. Please cite key studies or guidelines that establish this standard in the field.

·      Lines 57-59: Add appropriate references to support this statement. Providing evidence from previous research will strengthen the credibility of the argument.

·      Line 70: Please define "ICC" (Intraclass Correlation Coefficient) at its first use in the manuscript for clarity, especially for readers who may not be familiar with this term.

·      Line 74: The mention of “which results?” needs clarification. Specify which results you are referring to in order to avoid any confusion.

Materials and Methods

·      While the methods section includes some necessary components, its overall structure is unclear and could be better organized. Consider restructuring this section for better readability and flow.

·      The procedures are described in a somewhat disorganized manner. It would help to clearly delineate the steps of the study, using subsections where appropriate (e.g., participant recruitment, data collection, etc.).

·      The data analysis section is not appropriately placed within the methods. Please move this part to the relevant section of the manuscript for better coherence.

·      Overall, the methods section needs to be rewritten to improve clarity, structure, and completeness.

Results

·      The presentation of the results is currently unclear and difficult to follow. The findings should be summarized in a more concise and accessible manner. Consider breaking down complex results into simpler, more digestible points.

·      Additionally, before presenting the results, the methods section should clarify what exactly is being measured in the study. Providing a clearer outline of the variables and outcomes will help the reader better understand the results.

Discussion

·      The discussion section lacks structure and coherence. There seems to be little logical flow between paragraphs, which makes it difficult to follow the main argument. Consider reorganizing this section to create clearer transitions between ideas.

·      To improve the structure, it may help to base the discussion around the study’s hypotheses or main aims. This will allow for a more focused analysis of the findings and their implications.

Conclusions

·      The conclusions section is too lengthy for a typical journal article and reads more like a thesis. I recommend revising it to focus on the key findings of the study and their implications. This will make it more concise and accessible to the reader.

Comments on the Quality of English Language

I believe some minor improvements to the English in the manuscript could enhance its overall quality.

Author Response

Comment 1: Lines 28-29: The abstract would benefit from a brief mention of the data analysis methods used in the study. Including a few details (e.g., statistical tests or key analysis techniques) will provide a clearer understanding of the research approach.

Response: Thank you for your suggestion. We agree and have made the necessary changes in the revised manuscript.

Comment 2: Line 34: The term "AUC" is used only once in the abstract. I recommend removing this abbreviation, as it may not be necessary to define it in such a brief section.

Response: Thank you for pointing this out. We made the necessary changes in the revised manuscript.

Comment 3: The introduction does not provide enough background to fully grasp the relevance of the study. Specifically, there is insufficient explanation of why it is important to standardize and classify anterior cruciate ligament (ACL) injuries. More context is needed to help the reader

understand the significance of these issues in clinical practice.

Response: Thank you for your insightful comment. In the revised manuscript, we will expand the introduction to include more background on the importance of standardizing and classifying ACL injuries

Comment 4: Additionally, the introduction should describe the potential benefits of the app for

practitioners. How does it improve clinical decision-making or patient outcomes? This aspect

should be elaborated in more detail.

Response: Thank you for your valuable suggestion. We have revised the introduction to include a detailed discussion of the potential benefits of the PSM application for practitioners.

Comment 5: Line 50: When referring to the "gold standard," it is important to provide solid references to support this claim. Please cite key studies or guidelines that establish this standard in the field.

Response: We have updated the manuscript to include citations to key studies and guidelines that establish the pivot-shift test as a gold standard in the field of ACL diagnostics.

Comment 6: Lines 57-59: Add appropriate references to support this statement. Providing evidence from previous research will strengthen the credibility of the argument.

Response: We have revised the manuscript to include appropriate citations that highlight the challenges of feasibility, affordability, and comfort associated with devices such as arthrometers, robotic systems, and imaging modalities.

Comment 7: Line 70: Please define "ICC" (Intraclass Correlation Coefficient) at its first use in the manuscript for clarity, especially for readers who may not be familiar with this term.

Response: Thank you for your comment. We have revised the manuscript ensuring clarity.

Comment 8: Line 74: The mention of “which results?” needs clarification. Specify which results you are referring to in order to avoid any confusion.

Response: Response: Thank you for your comment. We have revised the manuscript (lines 110-126) ensuring clarity.

Comment 9: While the methods section includes some necessary components, its overall structure is unclear and could be better organized. Consider restructuring this section for better readability and flow.

Response: Thank you for your valuable feedback. We have restructured the Methods section to improve its organization, readability, and logical flow.

Comment 10: The procedures are described in a somewhat disorganized manner. It would help to clearly delineate the steps of the study, using subsections where appropriate (e.g., participant recruitment, data collection, etc.).

Response: We have revised the Methods section to clearly delineate the steps of the study, organizing them into subsections.

Comment 11: The data analysis section is not appropriately placed within the methods. Please move this part to the relevant section of the manuscript for better coherence.

Response: Thank you for your feedback. We have repositioned the data analysis section within the methods section to ensure better coherence and logical flow.

Comment 12: Overall, the methods section needs to be rewritten to improve clarity, structure, and completeness.

Response: We have rewritten this section to improve its clarity, structure, and completeness.

Comment 13: The presentation of the results is currently unclear and difficult to follow. The findings should be summarized in a more concise and accessible manner. Consider breaking down complex results into simpler, more digestible points.

Response: Thank you for your feedback. We have revised the Results section to present the findings in a more concise and accessible manner. The revised section is now organized into clearly defined subsections, and key findings are summarized using straightforward language

Comment 14: Additionally, before presenting the results, the methods section should clarify what exactly is being measured in the study. Providing a clearer outline of the variables and outcomes will help the reader better understand the results.

Response: We have revised the Methods section to provide a clearer outline of the variables and outcomes measured in the study.

Comment 15: The discussion section lacks structure and coherence. There seems to be little logical flow between paragraphs, which makes it difficult to follow the main argument. Consider reorganizing this section to create clearer transitions between ideas. To improve the structure, it may help to base the discussion around the study’s hypotheses or main aims. This will allow for a more focused analysis of the findings and their implications.

Response: Thank you for your thoughtful comment. We have reorganized this section to create a clearer logical flow.

Comment 16: The conclusions section is too lengthy for a typical journal article and reads more like a thesis. I recommend revising it to focus on the key findings of the study and their implications. This will make it more concise and accessible to the reader.

Response: Thank you for your observation. We have revised the Conclusions section to make it more concise and focused on the key findings and their implications.

Reviewer 4 Report

Comments and Suggestions for Authors

This manuscript entitled “Standardizing and Classifying Anterior Cruciate Ligament In-juries: International Multicenter Study Using a Mobile Application” was primarily aimed to assess the effectiveness of the Pivot-Shift Meter mobile application in diagnosing and classifying anterior cruciate ligament injuries. Authors bring an interesting study, but there are still some problems that cannot up this article to a publishing level. Suggestions are listed in the specific comments below.

Specific comments:

1.     In the Abstract part, line 22, “This international multicenter study is aimed to assess the effectiveness of …” Please replace “is aimed to” with “was aimed to”.

2.     In the Introduction part, I suggest that authors should provide more information about the knee ligament injuries. For example, the anatomy of the knee joint, ligaments around the knee joint, the prevalence of the knee ligament injuries, so on and so forth. This information could help us understand the importance of knee ligament injuries.

3.     In the Introduction part, line 47-48, “An injury to the ligament complex results in a loss of stability;” please cite the relevant paper to support this statement. Besides, can you be more specific about the stability here? Rotational stability? Or others?

4.     In the Introduction part, line 49-51, “Clinical evaluation includes tests such as the Lachman test, anterior drawer test, and Pivot-Shift (PS) test, considered the "gold standard" for assessing knee rotational function after an injury.” References are required here to support this statement.

5.     In the Introduction part, line 60, “…after ACL rupture” Please give the explanation of ACL in the article, since it first appeared in the article.

6.     In the Materials and Methods part, line 90-91, “A sample of patients with ACL injuries will be recruited, selected by trained Latin American medical specialists using the trial version of the PSM application.” Please write it in the past tense.

7.     For the Materials and Methods part, in the opinion of the reviewer, there is no adequate information for this study. Authors only provided selection criteria and procedure. Some information should be provided in the methods part instead of the results part. Besides, which machine learning algorithm did you use for classification? All this information should be provided in the methods part.

8.     In the Results part, the first 4 paragraphs are suitable for the methods part rather than the results.

9.     In the Results part, line 198-199, “with a significance level of 0.05, a p-value of .00001 was obtained,” Authors have also done the statistical analysis. If so, please add it in the methods part, and add the description about the significance level.

10.  Overall, the methods and results part are not well structured. Please consider rewriting these two parts.

11.  In the discussion part, it is recommended to provide a brief description of the aim and main findings in the first paragraph of the discussion part.

12.  In the discussion part, what are the limitations of this study? Please provide relevant description. Some recently studies could be added in the discussion, such as: Characteristics of Lower Limb Running-Related Injuries in Trail Runners: A Systematic Review’, Physical Activity and Health, 8(1), p. 137–147. Available at: https://doi.org/10.5334/paah.375.

13.  In the Conclusion part. In the opinion of the reviewer, the description in the conclusion part was too verbose, and the reviewer suggests that the authors should abbreviate the section and focus on the main findings of this study.

14.  Please do check the language and grammar mistakes throughout the whole article to further improve clarity.

Comments on the Quality of English Language

The English could be improved to more clearly express the research.

Author Response

Comment 1: In the Abstract part, line 22, “This international multicenter study is aimed to assess the effectiveness of …” Please replace “is aimed to” with “was aimed to”

Response: Thank you for pointing this out. The phrase in the Abstract has been revised as suggested.

Comment 2: In the Introduction part, I suggest that authors should provide more information about the knee ligament injuries. For example, the anatomy of the knee joint, ligaments around the knee joint, the prevalence of the knee ligament injuries, so on and so forth. This information could help us understand the importance of knee ligament injuries.

Response: Thank you for your suggestion. We have expanded the Introduction section to include additional information about knee ligament injuries

Comment 3: In the Introduction part, line 47-48, “An injury to the ligament complex results in a loss of stability;” please cite the relevant paper to support this statement. Besides, can you be more specific about the stability here? Rotational stability? Or others?

Response: Thank you for your insightful comment. We have clarified the type of stability being referred to in the sentence and specified that it pertains to rotational stability.

Comment 4: In the Introduction part, line 49-51, “Clinical evaluation includes tests such as the Lachman test, anterior drawer test, and Pivot-Shift (PS) test, considered the "gold standard" for assessing knee rotational function after an injury.” References are required here to support this statement.

Response: Thank you for your comment. We have added the relevant references to support the statement regarding the clinical evaluation of knee injuries, including the Lachman test, anterior drawer test, and Pivot-Shift test

Comment 5: In the Introduction part, line 60, “…after ACL rupture” Please give the explanation of ACL in the article, since it first appeared in the article.

Response: Thank you for pointing this out. We have added a brief explanation of the anterior cruciate ligament (ACL) when it first appears in the article.

Comment 6: In the Materials and Methods part, line 90-91, “A sample of patients with ACL injuries will be recruited, selected by trained Latin American medical specialists using the trial version of the PSM application.” Please write it in the past tense.

Response: Thank you for your comment. We have revised the Materials and Methods section and updated it.

Comment 7: For the Materials and Methods part, in the opinion of the reviewer, there is no adequate information for this study. Authors only provided selection criteria and procedure. Some information should be provided in the methods part instead of the results part. Besides, which machine learning algorithm did you use for classification? All this information should be

provided in the methods part.

Response: Thank you for your valuable feedback. We have expanded the Materials and Methods section to include additional details regarding the study design.

Comment 8: In the Results part, the first 4 paragraphs are suitable for the methods part rather than the results.

Response: Thank you for your helpful comment. We have revised the Results section to improve the clarity and organization of the manuscript.

Comment 9: In the Results part, line 198-199, “with a significance level of 0.05, a p-value of .00001 was obtained,” Authors have also done the statistical analysis. If so, please add it in the methods part, and add the description about the significance level.

Response: We have updated the Materials and Methods section to include a more detailed description of the statistical analysis performed in the study.

Comment 10: Overall, the methods and results part are not well structured. Please consider rewriting these two parts.

Response: We appreciate your suggestion and have revised the Materials and Methods and Results sections to improve their structure and clarity. The revisions focus on ensuring that both sections are well-organized and easier to follow, with a clear distinction between the methodology and the findings.

Comment 11: In the discussion part, it is recommended to provide a brief description of the aim and main findings in the first paragraph of the discussion part.

Response: We have revised the Discussion section to include a brief description of the aim of the study and the main findings.

Comment 12: In the discussion part, what are the limitations of this study? Please provide relevant description. Some recently studies could be added in the discussion, such as: ‘Characteristics of Lower Limb Running-Related Injuries in Trail Runners: A Systematic Review’, Physical Activity and Health, 8(1), p. 137–147. Available at: https://doi.org/10.5334/paah.375.

Response: Thank you for your constructive feedback. We have added a section discussing the limitations of the study and integrated the suggested reference into the Discussion section.

Comment 13: In the Conclusion part. In the opinion of the reviewer, the description in the conclusion part was too verbose, and the reviewer suggests that the authors should abbreviate the section and focus on the main findings of this study.

Response: Thank you for your feedback. We have revised the Conclusion section to make it more concise and focused on the main findings of the study.

Round 2

Reviewer 1 Report

Comments and Suggestions for Authors

thanks to the authors to response to my comments and modify the text

no more comments

Reviewer 3 Report

Comments and Suggestions for Authors

Dear authors,

After reviewing the revised manuscript, I must say that I still keep maintaining my initial decision to reject the submission. While you have made some changes in response to the previous feedback, I still find that the manuscript lacks sufficient empirical evidence to support its conclusions.

Comments on the Quality of English Language

N/A

Reviewer 4 Report

Comments and Suggestions for Authors

All my questions have been well addressed, I recommend to accept.